Meta-analysis of transcriptomic responses to biotic and abiotic stress in tomato

Ashrafi-Dehkordi Elham 1
Alemzadeh Abbas 1 alemzadeh@shirazu.ac.ir
Tanaka Nobukazu 2
Razi Hooman 1
1 Department of Crop Production and Plant Breeding, School of Agriculture, Shiraz University , Shiraz , Iran
2 Center of Gene Science, Hiroshima University , Kagamiyama, Higashi-Hiroshima, Hiroshima , Japan
Orlov Yuriy
Electronic publication date: 2018 Jul 17
Publication date: 2018
Volume: 6
Electronic Location ID: e4631
Received 2017 Sep 4; Accepted 2018 Mar 27
Copyright: © 2018 Ashrafi-Dehkordi et al.
Copyright year: 2018
Copyright holder: Ashrafi-Dehkordi et al.
License: This is an open access article distributed under the terms of the Creative Commons Attribution License, which permits unrestricted use, distribution, reproduction and adaptation in any medium and for any purpose provided that it is properly attributed. For attribution, the original author(s), title, publication source (PeerJ) and either DOI or URL of the article must be cited.
License URL: https://creativecommons.org/licenses/by/4.0/

Keywords: Abiotic and biotic stresses, Meta-analysis, Microarray, Solanum lycopersicum, Transcription factors

Funding: Iran National Science Foundation (INSF) and Shiraz University This work was supported by the Iran National Science Foundation (INSF) and Shiraz University. The funders had no role in study design, data collection and analysis, decision to publish, or preparation of the manuscript.

==============================
A wide range of biotic stresses (BS) and abiotic stresses (AS) adversely affect plant growth and productivity worldwide. The study of individual genes cannot be considered as an effective approach for the understanding of tolerance mechanisms, since these stresses are frequent and often in combination with each other, and a large number of genes are involved in these mechanisms. The availability of high-throughput genomic data has enabled the discovery of the role of transcription factors (TFs) in regulatory networks. A meta-analysis of BS and AS responses was performed by analyzing a total of 391 microarray samples from 23 different experiments and 2,336 differentially expressed genes (DEGs) involved in multiple stresses were identified. We identified 1,862 genes differentially regulated in response to BS was much greater than that regulated by AS, 835 genes, and found 15.4% or 361 DEGs with the conserved expression between AS and BS. The greatest percent of genes related to the cellular process (>76% genes), metabolic process (>76% genes) and response to stimulus (>50%). About 4.2% of genes involved in BS and AS responses belonged to the TF families. We identified several genes, which encode TFs that play an important role in AS and BS responses. These proteins included Jasmonate Ethylene Response Factor 1 (JERF1), SlGRAS6, MYB48, SlERF4, EIL2, protein LATE ELONGATED HYPOCOTYL (LHY), SlERF1, WRKY 26, basic leucine zipper TF, inducer of CBF expression 1-like, pti6, EIL3 and WRKY 11. Six of these proteins, JERF1, MYB48, protein LHY, EIL3, EIL2 and SlGRAS6, play central roles in these mechanisms. This research promoted a new approach to clarify the expression profiles of various genes under different conditions in plants, detected common genes from differentially regulated in response to these conditions and introduced them as candidate genes for improving plant tolerance through genetic engineering approach.

Introduction

As sessile organisms, plants constantly encounter a wide range of stresses that negatively affect the yield production, survival and growth of plants worldwide (Sharma et al., 2013). The stresses are divided into two broad categories, abiotic stresses (AS) including a variety of adverse environmental conditions, (i.e., drought, submergence, salinity, heavy metal contamination, nutrient deficiency, light and extreme temperatures) and biotic stresses (BS) caused by attack of living organisms, (i.e., bacteria, viruses, fungi and nematodes as well as other plants) (Rastgoo & Alemzadeh, 2011; Liu et al., 2014). To survive under stress conditions, complex mechanisms have evolved in plants to perceive signals from their environment, transmit through signal transduction pathways and respond to the various stresses that each of which may involve several hundred proteins (Fujita et al., 2006; Rastgoo, Alemzadeh & Afsharifar, 2011; Atkinson, Lilley & Urwin, 2013; Sharma et al., 2013; Sami & Alemzadeh, 2016). Functional molecular studies have figured out interactions between BS and AS signaling components with the identification a large number of genes, which are regulated by various types of transcription factors (TFs) (Kissoudis et al., 2016). TFs are proteins with critical roles in the control temporal and spatial gene expression during defense response that many of them are induced by more than one stress (Berrocal-Lobo, Molina & Solano, 2002; Liu et al., 2014; Shaik & Ramakrishna, 2014; Prasad et al., 2016).

The better understanding of plant stress response mechanisms under various stresses can give us a better view of how to improve the worldwide food production (Sharma et al., 2013). It is necessary to learn more about molecular mechanisms of TFs, MAPKs and ROS signal pathways, and hormone signaling, and the study of genes encoding regulatory molecules, especially TFs, can be more efficient than single-function genes, because TFs enable to regulate various genes to improve plant tolerance against stresses (Seki et al., 2002).

The bioinformatic approach is an emerging and rapidly developing field which allows researchers to analyze biological data (Khatri & Drăghici, 2005; Chang et al., 2013; Zinati, Alemzadeh & KayvanJoo, 2016). Recent advances in genome-wide transcriptome analysis methods, such as RNA sequencing and microarrays, enable researchers to simultaneously study the expression of thousands of genes and their co-expression partners under various stresses (Khatri & Drăghici, 2005; Sharma et al., 2013; Rest et al., 2016). Meta-analysis can be used to estimate differential gene expression between normal and stress conditions and find out some genes that their products are key molecules in response to the stress (Tseng, Ghosh & Feingold, 2012; Shaar-Moshe, Hübner & Peleg, 2015).

Today, with recent advances in high-throughput technology, a large number of biological data has become accessible via online resources, and we are now able to obtain results that more reliable by combining information from multiple sources. In this way, statistical methods can be used as useful and powerful tools to identify suitable candidate genes for test under experimental conditions to use in plant breeding programs. The combination of P-values is a common method in microarray meta-analysis and recently used in many studies (Tseng, Ghosh & Feingold, 2012). A meta-analysis of microarray using 13 experiments querying cytokinin-regulated gene expression changes in Arabidopsis combined with empirically defined filtering criteria identified a set of 226 genes differentially regulated by cytokinin (Bhargava et al., 2013). In another research project, a meta-analysis of 10 studies identified 4,015 genes showing significant differential expression in response to water limitation compared to control conditions in Arabidopsis (Rest et al., 2016).

Moreover, combining gene expression information across species can improve the ability to identify core gene sets with high evolutionary conservation. For example, a new cross-species meta-analysis of progressive drought stress at the reproductive stage was developed using Arabidopsis, rice, wheat and barley. Based on this dataset, important shared genes and metabolic pathways involved in whole plant adaptation to progressive drought stress across-species were identified (Shaar-Moshe, Hübner & Peleg, 2015). In this study, 225 differentially expressed genes (DEGs) shared across studies and taxa were identified and gene ontology (GO) enrichment and pathway analyses revealed that the shared genes were classified into functional categories involved predominantly in metabolic processes, regulatory function and response to stimulus (Shaar-Moshe, Hübner & Peleg, 2015).

Tomato (Solanum lycopersicum) is one of the most important food crops belongs to Solanaceae family, one of the largest and most important families of flowering plants. In addition, the tomato is a suitable model system to study environmental signal transduction components involved in BS and AS (Knapp et al., 2004; Arie et al., 2007). However, there is no a large-scale transcriptome analysis via meta-analysis of stress response studies in tomato to detect proteins, especially TFs, involved in various stresses.

In this study, for the first time we performed a large-scale comparative transcriptome analysis via meta-analysis of stress response studies in tomato using microarray gene expression data implemented on a single platform (Affymetrix Tomato Genome Array, Santa Clara, CA, USA). The goal of this meta-analysis was to detect DEGs involved in BS and AS and detect DEGs shared by both stresses. In addition, we focused on the identification of TFs have magnitude effects on the plant cells via up- and down-regulation of various genes under stress conditions. Other analysis methods, such as GO and Network analysis, were used to explain the complexity of tomato response to AS and BS. We also discussed the functions of the identified TFs in stress response.

Methods

Selection microarray studies

In this study, a meta-analysis of stress response was performed in tomato using publicly available microarray gene expression data implemented on a single platform (Affymetrix Tomato Genome Array, Santa Clara, CA, USA). Expression data of tomato plants exposed to BS and AS were combined. The raw expression data of various experiments were obtained from the Array Express from EBI (https://www.ebi.ac.uk/arrayexpress/).

Identification of differentially expressed genes

Microarray expression data from each source study was pre-processed separately as individual datasets. The raw data were normalized using Robust Multi-array Average background correction and quantile normalization. The data were processed in R version 3.2.2 (R Core Team, 2015). Meta-analysis of AS and BS was performed separately to identify DEGs involved in stress conditions. Genes that showed significant P-value (false discovery rate (FDR) of 5%), meaning that an estimated five percent of the DEGs are false positive, were considered as DEGs (Campain & Yang, 2010; Chang et al., 2013). Two methods of the combination of P-value have used: Fisher’s method and maxP (Wilkinson, 1951; Fisher, 1992). The Fisher’s combined probability test sums the logarithm-transformed P-values obtained from individual studies, under the null hypothesis, and follows a chi-squared distribution with 2k degrees of freedom, where k is the number of studies being combined (Rhodes et al., 2002). A very small P-value in just one study can be adequate to make statistical significance, even if the same genes are not significant in any other study. The maxP method takes maximum P-values across studies. It follows a beta distribution with parameters K (number of transcriptomic studies which are combined for a meta-analysis where each study contains G genes) and 1. This method targets on DEGs that have small P-values in all studies. (Chang et al., 2013; Song & Tseng, 2014).

Gene analysis

The genes selected by meta-analysis were further analyzed and characterized. In the first instance, the results were compared with the plant genes in the Plant Transcription factor & Protein Kinase Identifier and Classifier, iTAK (http://bioinfo.bti.cornell.edu/cgi-bin/itak/index.cgi) and the Plant Transcription Factor, PlantTFDB (http://plntfdb.bio.uni-potsdam.de/v3.0/) databases. The list includes 1,845 genes of tomato which encoding the TFs that directly or indirectly involved in signaling pathway and response to AS and BS. Venn’s diagrams (http://bioinfogp.cnb.csic.es/tools/venny/) were used to represent co-occurrence of DEGs. GO enrichment analysis of the DEGs (such as identifying biological processes) was conducted using the AgriGO platform (Du et al., 2010).

Network analysis

The Search Tool for the Retrieval of Interacting Genes/Proteins, STRING 10.5 (http://www.mybiosoftware.com/string-9-0-search-tool-retrieval-interacting-genesproteins.html) (Szklarczyk et al., 2014) database was used to figure out all functional relations between the genes and their occurrence patterns across various genomes. The database makes relations based on several lines of evidences: the empirical evidence from protein–protein interaction assays, co-expression data from the NCBI Gene Expression Omnibus database, the extraction of information from other databases, coexistence of the genes in the same organisms, conserved gene neighborhood in known genomes, gene fusion events, pathway annotation from other resources such as the Kyoto Encyclopedia of Genes and Genomes or GO databases, and automated text-mining tools (Szklarczyk et al., 2014). STRING calculates a confidence value for those interactions according to the pieces of evidences from above 0.4 to 0.9, as the medium to highest score, respectively.

Results and Discussion

Identification of genes involved in biotic and abiotic stresses

After searching in the database, microarray data from 23 different experiments were selected and analyzed. To obtain a global analysis, 391 microarray samples including 213 AS from different categories: drought, heat, salinity, light, hormone, mineral and heavy metals deficiencies/toxicities and 178 BS that caused by viruses, nematode, fungi, bacteria and pests in tomato (wild type and transgenic) were chosen (Table S1). We found a total of 1,862 genes differentially regulated in response to BS, was far greater than that induced by AS, 835 genes, with a FDR ≤ 0.05 by Fisher’s statistical method (Tables S1–S3). We also figured out 15.4% or 361 DEGs with the conserved expression between AS and BS by Fisher’s statistical method, suggesting that these genes and their associated cell-signaling pathways are regulated in a similar way in a wide range of stresses (Fig. 1A; Table S5). Using maxP method, a total of 14 genes differentially regulated in response to AS and a total of 220 genes differentially regulated in response to BS were found. We figured out 0.9% or 2 DEGs (LES.1399.1 and LES.3969.1) with the conserved expression between AS and BS by maxP method (Fig. 1A; Tables S2–S4). These methods used in other studies to identify important genes in different physiological processes in other plants. Shaar-Moshe, Hübner & Peleg (2015) identified 225 DEGs across-species by using 17 microarray experiments of drought stress at the reproductive stage of Arabidopsis, rice, wheat and barley. Fisher’s and maxP methods are two popular microarray meta-analysis methods used in the literature that have different strength for detecting different types of DEGs (Tseng, Ghosh & Feingold, 2012; Wang et al., 2012). Our results also showed that the two used methods detected different sets of DEGs, suggesting different assumptions behind the methods. Among these methods, Fisher’s method is more popular in the microarray meta-analysis studies (Wang et al., 2012); hence, we were more interested in detecting DEGs across all studies through Fisher’s statistical method.

Figure 1 Identification of genes involved in biotic and abiotic stresses.

Comparison of differentially expressed genes (DEGs) under abiotic and biotic stress responses. (A) Four-way Venn diagrams showing co-occurrence of DEGs in response to various abiotic and biotic stresses by two meta-analytical approaches: Fisher and maxP methods. (B) Four-way Venn diagrams showing number of transcription factors DEGs in all identified abiotic stresses by two different meta-analytical statistical methods: Fisher and maxP. (C) Four-way Venn diagrams showing number of transcription factors DEGs in all identified biotic stresses by two different meta-analytical statistical methods: Fisher and maxP.

Gene ontology

We focused on the 361 DEGs common to both AS and BS and subjected them to GO and network analysis to explore other possible functions of them. GO enrichment analysis of the DEGs showed that shared DEGs represented genes involved in main biological and cellular processes. Among them, the greatest percentage of genes involved in the cellular process (>76% genes), metabolic process (>76% genes) and response to stimulus (>50%). One of largest functional group of DEGs was associated with metabolic processes (e.g., carbohydrate metabolic process, lipid metabolic process, protein metabolic process, nucleobase, nucleoside, nucleotide and nucleic acid metabolic process, cellular amino acid and derivative metabolic process and regulation of primary metabolic process).

In addition, another large functional group of DEGs was associated with cellular processes (e.g., cellular metabolic process, regulation of cellular process and cellular response to stimulus), the show a significant rearrangement in plant metabolism as part of stress adaptation (Shaar-Moshe, Hübner & Peleg, 2015).

On the other hand, the lowest percentage of genes involved in positive regulation of biological process (5%) (Fig. 2; Table S6). The most significant GO terms for cellular process were cellular metabolic process (FDR: 1.8E-37), organic acid metabolic process (FDR: 9.2E-29), cellular ketone metabolic process (FDR: 1.2E-28) in metabolic process were primary metabolic process (FDR: 7.8E-39), cellular metabolic process (FDR: 1.8E-37) and organic acid metabolic process (FDR: 9.2E-29). Moreover, in response to stimulus were response to abiotic stimulus (FDR: 3.4E-40), response to biotic stimulus (FDR: 9.4E-26), and response to stress (FDR: 9.4E-26). Some of these genes possess important roles in DEGs shared by AS and BS networks (Table S7). A genome-wide transcriptional analysis was performed just for Fe deficiency and was identified 97 DEGs by comparing Fe-deficient and Fe-sufficient in roots of tomato. These transcripts are related to the physiological responses to the nutrient stress resulting in an improved iron uptake, including regulatory aspects, translocation, root morphological modification, and adaptation in primary metabolic pathways, such as glycolysis and TCA cycle (Zamboni et al., 2012). Another research project showed that the largest functional groups of DEGs was associated respectively to metabolic processes (e.g., amino acid and carbohydrate metabolism), regulator function (e.g., protein degradation and transcription) and response to the stimulus that was identified by the research of Shaar-Moshe, Hübner & Peleg (2015).

Figure 2 Gene ontology analysis.

Frequency of most representative biological process terms. Com: Differentially expressed genes obtained by common genes in biotic and abiotic stresses, AS: Differentially expressed genes obtained by abiotic stresses, BS: Differentially expressed genes obtained by biotic stresses. BG: The frequency of these terms in the reference genes, Tomato Locus set. Gene Ontology analysis made in the AgriGO platform (FDR = 5%). More details in Tables S10 and S11.

Network analysis

Network analysis using STRING 10.5 identified the connections among DEGs common to AS and BS (Fig. 3; Table S7). In addition, some genes that possess an important role in the three biological pathways; cellular process, metabolic process, response to stimulus were detected. For example, Solyc02g090890.2.1 (ZE: plays an important role in resistance to stresses, seed development and dormancy), Solyc04g049350.2.1 (CS1: biosynthesis of aromatic amino acids), Solyc07g052480.2.1 (LOC544276: involved in storage lipid mobilization during the growth of higher plant seedling), Solyc07g049530.2.1 (ACO1: involved in ethylene biosynthesis via S-adenosyl-L-methionine), Solyc01g099630.2.1 (XTH1: an essential component of the primary cell wall, and thereby participates in cell wall construction of growing tissues), Solyc05g055230.1.1 (RPS17: ribosomal small subunit assembly), Solyc12g098350.1.1 (infA: no specific function has so far been attributed to this initiation factor; however, it seems to stimulate more or less all the activities of the other two initiation factors, IF-2 and IF-3), Solyc02g082920.2.1 (CHI3: defense against chitin containing fungal pathogens), Solyc06g082600.2.1 (UBC3: mediates the selective degradation of abnormal and short-lived proteins), Solyc01g096480.2.1 (LOC543657: probable thiol-disulfide oxidoreductase that may play a role in proper chloroplast development and negatively regulates Cf-9/Avr9-mediated cell death and defense responses), Solyc07g047850.2.1 (CAB4: it receives and transfers excitation energy to photosystems).

Figure 3 Network analysis of the DEGs identified in biotic and abiotic stresses.

Network analysis of the 361 common DEGs identified in biotic and abiotic stresses. Network was generated by STRING (version 10.5) database and represents all connections of those genes with a confidence score > 0.4. The connection colors show the types of evidence for concluding association: co-occurrence of those genes in the same organisms (dark blue), co-expression (black), experimental protein–protein interaction data (pink) and literature text-mining (yellow). A number above hub represent TF. (1) JERF1, (2) MYB48, (3) protein LHY, (4) EIL3, (5) EIL2, (6) WRKY26.

The results of network analysis also showed some of the TFs have connections to other molecules. Based on pieces of literatures reviewing, detected TFs in this study, Jasmonate Ethylene Response Factor 1 (JERF1), EIL3, EIL2, SlGRAS6, Protein LATE ELONGATED HYPOCOTYL (LHY) and MYB 48 affected by a wide range of stresses and possibly play essential and significant roles in multiple stress responses which is in agreement with their functional roles in BS and AS responses. Expression of JERF1 was induced by salt, ethylene, MeJA and abscisic acid (ABA) treatments in tomato that may be pointed to a central role for JERF1 in different signal transduction pathways (Zhang et al., 2004). It has been also reported that JERF1 activates ABA biosynthesis related genes such as NtSDR (short-chain dehydrogenase/reductase) and enhanced cold and salt stress tolerance in transgenic tobacco (Wu et al., 2007; Ouyang et al., 2007). It showed that the expression of EIL2 and EIL3 was induced by ethylene and salinity treatments (Ying et al., 2004) and SlGRAS6 is one of the most important members of the tomato GRAS family in disease resistance and mechanical stress (Mayrose et al., 2006). Different studies have shown the large family of MYB has essential roles in plant growth, development, primary and secondary metabolism and response to BS and AS (Oh, Park & Han, 2003; Van der Ent et al., 2008; Dou et al., 2016). It was also reported that LHY controls the expression of a large number of ABA-responsive genes by binding directly to their promoters (Huang et al., 2007). These results support that the detected TFs in our study might play an important role in BS and AS responses.

Identification of transcription factors involved in biotic and abiotic stresses

In this research, because of the importance of TFs as a powerful tool for the manipulation of complex metabolic pathways, we have tried to identify genes encoding them. According to the PlantTFDB, 1,845 genes from tomato encoding TFs were identified and classified into 58 families that directly or indirectly involved in signaling and response to AS and BS (Table S8). Based on our results, from the 1,862 genes response to BS selected by Fisher’s statistical method, 93 genes (5%) encode TFs (Fig. 1C; Table S9). Ethylene-responsive transcription factors (ERF) family has the greatest number of genes (18 genes), and C3H, FAR1, GATA, LBD, MIKC-MADS, PHD, SLP protein, TALE, TIFE and YABBY families have the lowest number of genes (each family has only one gene) (Table 1). From the 835 genes response to AS selected by Fisher’s statistical method, 31 of those (3.7%) encode TFs (Fig. 1B; Table S10). ERF and NAC families have the greatest number of genes (five genes), and ARF, BBR-BPC, GRAS, NF-NY and VOZ families have the lowest number of genes (each family has only one gene) (Table 1).

Table 1 Transcription factors and their binding sites.

TF family	No. of TF in abiotic	No. of TF in biotic	TFBS	
ARF	1	3	TGTCTC auxin response elements (AuxRE)	
BBR-BPC	1	–	(GA/TC)8 and GAGA-binding	
bHLH	3	9	E-box (5′-CANNTG-3′)	
C2H2	–	9	A DNA element that contains an AGCT core	
C3H	–	1	Unknown	
ERF	5	18	GCC box is an 11 bp sequence (TAAGAGCCGCC)	
EIL	3	2	EIL2 BS in ERF1 (TTCAAGGGGGCATGTATCTTGAA)	
FAR1	–	1	FBS for FHY3-FAR1 binding site	
GATA	–	1	WGATAR (W = T or A; R = G or A) motifs	
GRAS	1	7	cis-element AATTT	
HSF	–	3	This consists of a tandem of inverted repeats of the sequence GAA, generating a perfect HSE, TTCnnGAAnnTTC	
LBD	–	1	Core sequence CGGC	
MIKC_MADS	–	1	Keratin-like coiled-coil domain	
MYB and MYB_related	2	8	(CNGTTR), (GKTWGTTR), TAACPy sequence (only one AAC sequence) and (GKTWGGTR; R, A or G; K, G or T; W, A or T) and EE, AGATATTT	
NAC	5	4	NAC recognition site (NACRS), NAC binding element (SNBE) with a longer and variable sequence ([T/A]NN[C/T][T/C/G]TNNNNNNNA[A/C]GN[A/C/T][A/T])	
NF-YB	–	3	CCAAT binding	
NF-YC	1	–	Core nucleotide sequence CCAAT	
PHD	–	1	Unknown	
SPL proteins	–	1	SBP-box, TNCGTACAA	
TALE	2	1	Pbx:Meinox binding site (CTGTCAATCA)	
TCP	–	2	GGNCCCAC sequences and s G(T/C)GGNCCC	
TIFY	–	1	Unknown	
VOZ	1	–	GCGTNx7ACGC	
WRKY	3	9	W box (TTGACY; Y, C or T)	
YABBY	–	1	WATNATW (W = T or A; R = G or A)	
ZIP and HD-ZIP	3	6	Motif AATNATT	
Notes:

Transcription factor and binding site for each, in biotic and abiotic stresses. TF, Transcription factor; TFBS, Transcription factor binding site.

Since TFs are the key regulators of plant growth, development and metabolisms via up- and down-regulation of various genes, identification of the genes encoding these proteins and transferring them into plants to improve their stress tolerance would be interesting. These TFs (JERF1, Pti6, SlERF1, SlERF4, SlGRAS6, MYB48, protein LHY, EIL2, EIL3, WRKY 11, WRKY 26, inducer of CBF expression 1 (ICE1)-like and bZIP TF) can be used as candidate genes in molecular breeding programs because these TFs are common to AS and BS; also, our results indicate they are being affected by a wide range of stresses. In addition, they possibly play essential and significant roles in multiple stress responses, which is in agreement with their functional roles in BS and AS responses (Gu et al., 2002; Ying et al., 2004; Zhang et al., 2004, 2010; Borrás-Hidalgo et al., 2006; Mayrose et al., 2006; Wu et al., 2007; Yáñez et al., 2009; Sharma et al., 2010; Chen et al., 2012; Kim, Stork & Mudgett, 2013; Feng et al., 2013; Li, Luan & Liu, 2015). The results of network analysis also showed some of these proteins have central roles in BS and AS responses (Fig. 3). The two-way Venn diagram indicated that the number of common TFs between BS and AS was 13 while 18 and 80 TFs were uniquely identified in AS and BS, respectively (Fig. 4).

Figure 4 Common transcription factors in biotic and abiotic stresses.

Two-way Venn diagram showing the common transcription factors between abiotic and biotic stresses.

ERF family

The ERF has been shown to play a key role during plant development and stress tolerance mechanisms. Overexpression of ERF family genes in different plant species such as Arabidopsis, tobacco, rice and tomato has been shown to increase tolerance to a wide range of BS and AS (Sharma et al., 2010). The tomato has 112 AP2/ERF superfamily genes (Sharma et al., 2010, 2013). Our results showed that four of 13 TFs shared between AS and BS belonged to ERF family, including JERF1, SlERF4, SlERF1 and pti6 (Table 2).

Table 2 Detected transcription factors.

Transcription factor family	Transcription factor	Function	References	
ERF	JERF1	Salinity, low temperature, drought and various biotic stresses	Zhang et al. (2004), Wu et al. (2007), Sharma et al. (2010), Zhang et al. (2010)	
	SlERF4	Abiotic stress and pathogen (Xanthomonas campestris)	Sharma et al. (2010), Kim, Stork & Mudgett (2013)	
	Pti6	Pathogen	Gu et al. (2002), Chakravarthy et al. (2003), Sharma et al. (2010)	
	SlERF1	Abiotic stress and pathogen (Plectosphaerella cucumerina and Botrytis cinerea)	Gu et al. (2002), Mayrose et al. (2006)	
GRAS	SlGRAS6	Abiotic stress, mechanical stress and pathogen (Pseudomonas syringae pv. tomato)	Mayrose et al. (2006)	
MYB family and MYB-related family	MYB48	Salt, drought, cold and pathogen	Xiong et al. (2014)	
	LHY	Low temperature and pathogen	Huang et al. (2007), Alves et al. (2013), Grundy, Stoker & Carré (2015), Seo & Mas (2015)	
EIL	EIL2	Salinity and ortholog in tobacco resistance against pathogene (Peronospora hyoscyami)	Ying et al. (2004), Borrás-Hidalgo et al. (2006)	
	EIL3	Salinity and ortholog in tobacco resistance against pathogene (Peronospora hyoscyami)	Ying et al. (2004), Borrás-Hidalgo et al. (2006)	
WRKY	WRKY11	Drought, heat tolerance and pathogen	Wu et al. (2009), Li et al. (2011), Chen et al. (2012), Alves et al. (2013)	
	WRKY26	Heat stress and pathogen	Li et al. (2011), Chen et al. (2012), Alves et al. (2013)	
ICE1-like bHLH	ICE1-like	Cold, chilling, osmotic and salt	Chinnusamy et al. (2003), Feng et al. (2013), Huang et al. (2015)	
bZIP	bZIP11	Abiotic stress and pathogen	Hummel et al. (2009), Alves et al. (2013)	
Note:

Detected transcription factors are affected by a diverse set of biotic and abiotic stresses.

Jasmonate Ethylene Response Factor 1, has critical roles in plant abiotic stress responses, such as salinity, low temperature, dehydration and various BS in different plant species (Zhang et al., 2004, 2010; Wu et al., 2007). It has been shown that the expression of JERF1 in tomato was induced by salt, ethylene, MeJA and ABA treatments that may be pointed to a central role for JERF1 in different signal transduction pathways (Zhang et al., 2004). Overexpression of JERF1 enhanced the tolerance of transgenic tobacco and rice plants to high salinity, low temperature, osmotic stress and pathogen attack by regulating the expression of downstream stress-responsive genes and a number of pathogenesis-related (PR) genes due to binding to DRE/CRT and GCC-box cis-elements (Wu et al., 2007; Zhang et al., 2010). Expressing JERF1 in tobacco induced expression of GCC box-containing genes such as CHN50, Prb-1b, GLA, NtSDR and osmotin that subsequently resulted in enhanced tolerance to salt stress and low temperature (Zhang et al., 2004; Wu et al., 2007). JERF1 up-regulated the expression of genes related to osmolyte synthesis such as OsP5CS and OsSPDS2, which, has increased the production of osmolytes. It also induced the expression of OsABA2 and Os03g0810800 genes (containing DRE element in their promoter regions), which encode two key enzymes involved in ABA biosynthesis and increased the synthesis of ABA in transgenic rice that resulted in elevated tolerance to drought (Zhang et al., 2010). JERF1 may also be located downstream of EIN3 in the ethylene signaling pathway (Zhang et al., 2004). It has been previously reported that SlERF4, an ERF in tomato, is a XopD, a type III secretion effector from Xanthomonas campestris, destabilize SlERF4 and this task has been proposed to promote bacterial growth (Kim, Stork & Mudgett, 2013). SlERF4 reduces ethylene production and, in turn, limits disease symptom development (Kim, Stork & Mudgett, 2013) (Table 2). Pti6 belonged to the ERF family in the plant that binds specifically to the GCC-box cis-element present in the promoter regions of many PR genes and induced the expression of a wide range of these genes that have key roles in the plant defense. It was shown that the expression of jasmonic acid- and ethylene-regulated genes, such as PR3, PR4, PDF1.2 and Thi2.1, was being differently affected by Pti4 or other similar genes, Pti5, or Pti6 (Gu et al., 2002; Chakravarthy et al., 2003) (Table 2). The gene encoding SlERF1 was another detected gene that probably has an important role in response to various stresses. It has been shown that the overexpression of SlERF1 in Arabidopsis can confer resistance to necrotrophic fungi like Plectosphaerella cucumerina and Botrytis cinerea (Mayrose et al., 2006) (Table 2).

GRAS family

SlGRAS6 was another identified TF that is one of the most important members of tomato GRAS family in disease resistance and mechanical stress (Mayrose et al., 2006). SlGRAS6 is required for tomato disease resistance to the bacterial pathogen Pseudomonas syringae pv. tomato. It has been shown that the suppression of SlGRAS6 gene expression through virus-induced gene silencing reduced resistance of tomato to Pseudomonas syringae pv. tomato (Mayrose et al., 2006) (Table 2).

MYB family and MYB-related family

The MYB is a large TF family in plants that have essential roles in plant growth, development, primary and secondary metabolism and response to BS and AS (Oh, Park & Han, 2003; Van der Ent et al., 2008; Dou et al., 2016). Most of MYB genes involved in response to various AS, belong to the R2R3-type group. With the comparison of OsMYB48-1 expression with other MYB proteins, a special role for this gene in response to the AS in rice can be suggested. It has been shown that its expression was strongly induced by ABA, H2O2, PEG, and dehydration stresses, while slightly induced by salt and cold stresses (Xiong et al., 2014). It was reported that the overexpression of OsMYB48-1 regulated the expression level of some stress responsive-genes, such as RAB21, RAB16D, RAB16C and LEA3, under drought stress conditions (Xiong et al., 2014) (Table 2). The LHY TF plays a critical role in the regulation of low temperature stress response via controlling expression of CBF 1, 2 and 3, COR27 and COL1 genes and contribute to plant defense through the circadian regulation of stomatal aperture (Grundy, Stoker & Carré, 2015; Seo & Mas, 2015). It was also shown that LHY controls the expression of a large number of ABA-responsive genes by binding directly to their promoters (Huang et al., 2007).

EIL family

It has been shown that the expression of EIL2 and EIL3, which encode TFs in ethylene signal transduction, was induced by ethylene and salinity treatments (Ying et al., 2004). It also reported that its ortholog in tobacco is required for resistance against Peronospora hyoscyami (Borrás-Hidalgo et al., 2006) (Table 2).

WRKY family

Different studies have shown that the members of WRKY family contribute multiple biological processes such as plant growth and development. They also have important roles in a wide range of BS and AS such as drought, salt, invasion of pathogens, implying that the members of this family may be putative regulators in response to various BS and AS (Huang et al., 2012; Li, Luan & Jin, 2012). Two of 13 TFs, WRKY11 and WRKY26, which are involved in both AS and BS, belong to the WRKY family (Table 2). In other plant species, it was also shown that the members of this family have a key role in AS and BS responses (Li et al., 2011; Chen et al., 2012). It was reported that OsWRKY11 involved in drought and heat tolerance in rice (Wu et al., 2009). It was shown that, in Arabidopsis thaliana, WRKY26 positively regulates the cooperation between heat shock proteins and ethylene-activated signaling pathways that mediate responses to heat stress (Li et al., 2011). It has also been shown that the promoters of some heat tolerance-related genes including Hsp and Hsf genes such as HsfA2, HsfB1, Hsp101, and MBF1c contain W-box sequences that are recognized by WRKY proteins (Li et al., 2011).

ICE1-like bHLH family

Inducer of CBF expression 1-like encodes a MYC-type basic helix-loop-helix (bHLH) TF (Chinnusamy et al., 2003). ICE1 is an important inducer of CBF3/DREB1A which regulates cold stress tolerance (Chinnusamy et al., 2003; Huang et al., 2015; Feng et al., 2013) (Table 2). In Arabidopsis, ice1 mutants showed decreased cold and chilling tolerance, whereas ICE1 overexpression in Arabidopsis increased the cold tolerance of transgenic plants (Chinnusamy et al., 2003). Furthermore, osmotic and salt tolerance were also much markedly increased in transgenic tobacco carrying SlICE1 from tomato by increasing the expression level of DREB1/CBF and their target genes (Feng et al., 2013) (Table 2).

bZIP family

Transcription factors of the bZIP family regulate various biological processes in plant growth and development, including morphogenesis and seed formation, as well as in AS and BS responses by binding to promoters of specific target genes to up- or down-regulate their expression (Li et al., 2015). WRKY TFs, bZIP domain and MYB TFs involved in plant defense against pathogens (Alves et al., 2013) (Table 2). TF bZIP11 is a member of the S1 class of the bZIP family, and it was shown that bZIP11 mRNA translation is repressed in response to sucrose (Hummel et al., 2009). It has been shown that sucrose repression of bZIP11 mRNA translation depends on the presence of the 5′-leader sequence upstream of bZIP11 gene (Hummel et al., 2009). It has also been shown that sucrose signaling has an important role in the translational control of bZIP11 in response to stress conditions (Hummel et al., 2009).

These reports implied that these TFs are important regulators of biological processes and may be useful in the breeding of plants to improve their tolerance against adverse stresses. The presence or absence of the TFs across various plant species and other organisms including bacteria, archaea and other eukaryotes is shown in Fig. 5. A high homology was observed for all TFs except protein LHY between S. lycopersicum and S. tuberosum and the highest percentage of similarity was found between Vitis vinifera, Populus trichocarpa and Glycine max. The results showed that GRAS6 is the only TF present in bacteria with a low similarity to that of tomato GRAS6. Just three of the identified TFs, ICE1-like, protein LHY and MYB48, are present in Opisthokonta that the homology between MYB48 and those of Opisthokonta was the highest. MYB48 is the only protein, which is present in all organisms except bacteria and archaea (Fig. 5). It has been shown that MYB is one of the largest and most diverse TF families, which are the key factors in regulatory networks controlling metabolism, development and responses to BS and AS (Roy, 2016). The members of MYB family found in all eukaryotes and molecular studies in Arabidopsis has been shown the role of this family in the epigenetic regulation of stress responses in plants (Roy, 2016). The results showed MYB48 as one of the members of MYB family possesses an important role in the network of abiotic and biotic responses (Fig. 3). In addition to MYB48, three other TFs, WRKY11, WRKY26 and protein LHY, were observed in Amoebozoa with low similarity to those of tomato (Fig. 5). As well as Amoebozoa, the WRKYs were observed in Giardia lamblia, also known as Giardia intestinalis (Fig. 5). It was previously thought that WRKYs are specific to plants and algae (Eulgem & Somssich, 2007) but recent studies have shown their presence in human protozoan parasite G. lamblia and Dictyostelium discoideum (slime mold) (Wu et al., 2005). It has been reported that WRKYs also have important roles in other physiological processes. It was shown that the expression of genes encoding proteins of cell wall increased, when a WRKY-like gene overexpressed in G. lamblia (Pan et al., 2009).

Figure 5 The presence or absence of the transcription factors across various organisms.

Transcription factor occurrence patterns across various genomes. (1) EIL2, (2) EIL3, (3) ZIP, (4) GRAS6, (5) SlERF4, (6) SlERF1, (7) JERF1, (8) WRKY26, (9) ICE1-like, (10) pti6, (11) WRKY11, (12) protein LHY, (13) MYB48. The intensity of the color of the red square reflects the amount of conservation of the homologous protein in the species. White color (No similarity detectable) and black/red color (100 similarity detectable).

The results showed that protein LHY present in all eukaryotes except G. lamblia, Trichomonas vaginalis and Trypanosomatidea (Fig. 5) and also suggested a central role for this protein in the network of abiotic and biotic responses (Fig. 3). The results of other studies also support an important role for this protein and suggested that this protein involved in circadian clock control (Lu et al., 2009).

Conclusion

Transcriptome-wide studies by meta-analysis allowed us to narrow down DEGs to a small number. DEGs represent the genes in different networks and signaling pathways that enable us to figure out which genes play the major role in various stresses and determine the overlap between BS and AS response pathways.

This study was designed to identify genes, which, have important roles in abiotic and biotic tolerance mechanisms in tomato and focused on those shared by both stresses. The novelty of the work presented here is that a large meta-analysis was used to identify important genes in tomato as a model of Solanaceae crops. Meta-analysis has been previously used to figure out important genes that play critical roles in response to BS or AS in other plants such as Arabidopsis (Shaar-Moshe, Hübner & Peleg, 2015; Rest et al., 2016). However, this is the first time that the tomato was selected as the target plant to detect genes involved in a wide range of BS and AS responses. In previous studies, usually one or a few stresses were considered, but here a wide range of stresses was studied simultaneously. We identified 1,862 and 835 genes differentially regulated in response to BS and AS, respectively, and 361 genes identified in this work are to be involved in cross-talk between BS and AS responses. Since TFs regulate the expression of a wide range of genes under stress conditions, we focused on genes which encode TFs and it was revealed that close to 5% of identified genes encoding TFs. Among them, the expression of genes encoding JERF1, GRAS6, MYB48, SlERF4, EIL2, protein LHY, SlERF1, WRKY 26, bZIP TF, ICE1-like, pti6, EIL3 and WRKY 11 are being affected by a diverse set of BS and AS. This study highlighted the importance of detected TFs as regulators of various stress responses, reinforcing the role of these TFs in regulating plant tolerance to a board spectrum of stresses (Borrás-Hidalgo et al., 2006; Yáñez et al., 2009; Sharma et al., 2010; Feng et al., 2013; Li, Luan & Liu, 2015). The results suggest that detected TFs in this work play important roles in plant tolerance response to various stresses, and the data generated will be helpful in conducting functional genomics studies to figure out their precise role during plant stress response.

The salient features of this study include: (1) a valuable approach to identify the expression profiles of various genes under different conditions in plants; (2) meta-analysis that can be used to characterize candidate genes for both BS and AS tolerance; (3) identification of important genes involved in various BS and AS; and (4) identified genes which are potential candidates for the genetic manipulation of plants to improve their tolerance to BS and AS.

Supplemental Information

Supplemental Information 1 Supplementary material.

Raw data and more analysis of data.

Click here for additional data file.

The authors are thankful to Dr. Ahmad Tahmasebi for his invaluable assistance throughout this study.

Additional Information and Declarations

Competing Interests

Author Contributions

Data Availability

The authors declare that they have no competing interests.

Elham Ashrafi-Dehkordi performed the experiments, analyzed the data, prepared figures and/or tables.

Abbas Alemzadeh conceived and designed the experiments, authored or reviewed drafts of the paper, approved the final draft.

Nobukazu Tanaka authored or reviewed drafts of the paper.

Hooman Razi analyzed the data.

The following information was supplied regarding data availability:

The raw data are provided in the Supplemental Files.

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
