# Peer review of "Meta-analysis of transcriptomic responses to biotic and abiotic stress in tomato"

_PeerJ, doi:10.7717/peerj.4631_

## Round 0.1 · original submission · Major Revisions

The paper highlights actual problem of stress-resistance studies in plants by bioinformatics methods. Though 3 reviewers recommended revision, the main remarks are on English and presentation style. Please correct accordingly, taking care on statistics description (validation). I await your revised manuscript.

Reviewer 1 ·

Basic reporting

This manuscript requires major English language copyediting. The abstract requires major overhaul as with regards to using the English language professionally.

In Figure 5, the font size is too small to read the various organisms listed.

Figure 3 seems to be very crowded and hard to comprehend.

The authors should also gather more information for review of related literature related to this study. There seems to be a number of studies related to this one.

Experimental design

The authors should be clear in the goals of this study. They should also be able to clearly express the novelty and significance of this study in the introduction.

Validity of the findings

It is unclear as to how this study was validated. Did the authors provide training and validation sets to verify the findings? For the training set, how were the results cross-validated?

Reviewer 2 ·

Basic reporting

The writing quality is somewhat subpar and the manuscript has grammatical errors that are too numerous to address individually. This hampers the flow of the paper and makes it difficult to understand key results and insights from the study.

Literature references are sufficient and an appropriate background is provided

Experimental design

The research aims are well-defined and are relevant. In this study, the authors use meta-analytic statistical techniques and bioinformatics tools to identify and characterize genes in tomato that are differentially expressed during responses to stress conditions. The analysis is based on publically available microarray tomato gene expression data. The analysis meets the appropriate level of rigor and the results are laid out in sufficient detail.

Validity of the findings

The data is robust and the analytical technique is statistically sound. However, the conclusion is stated in poor English that is difficult to understand. The Conclusion section is minimal, and the authors do not provide a discussion about how their findings connect with the study aims and how their work fits into the larger framework of the related literature. The manuscripts reads too much like a report

Additional comments

The Results section is not well organized and difficult to read. Also, it includes a lot of material that is better suited for the Discussion/Conclusion section. The reviewer suggests summarizing the findings in the Results in 1 or 2 tables, so as to provide a compact summary that readers can easily understand. The reviewer also suggests expanding the Conclusion section to discuss the implications of the findings.

Below are some specific suggestions:

Page 5, Line 95: The sentence that begins with "Our analysis revealed...." is out of place here. Do not state results in the introduction section. This should be moved to the Results section.
Page 6, Lines 98-100: this sentence is unclear, please reword.
Page 6, Line 115: Citations should be given for Fisher's method and maxP
Page 7, Line 120: The authors state the maxP statistic follows a beta distribution with K degrees of freedom, what does K represent? Also, the authors state beta=1 under the null, but don't define what beta is. Is beta a parameter of the beta distribution? If so, this needs to be stated clearly
Page 7 line 121: "A gene to be detected by this method if has small P-values in all studies", please correct the grammatical errors so that it is clear to readers what the procedure is.
Page 7 line 136: the sentence is ended prematurely, "Database" should not be capitalized
Page 8 lines 151-155: Only the results using Fisher's statistical method are stated, but not maxP results
Page 10 lines 197: The sentence beginning with "Forasmuch as,..." should be reworded correctly
Page 16/17 lines 346-348: this statement of this result is critical but it is incorrectly worded. It is difficult to understand what is being stated here

Reviewer 3 ·

Basic reporting

The work is written in fairly good English, excluding ambiguous interpretation. Terms are used correctly and to the place.

References to the literature are sufficient to understand the relevance of the key problem. However, in my opinion, the authors overly briefly cite the work on meta analysis of expression data. I recommend improving the fragment of 77-82 with sufficient details. Meta transcriptome analysis in various variations is actively used for plants before, for example:

Rest, J. S., Wilkins, O., Yuan, W., Purugganan, M. D., & Gurevitch, J. (2016). Meta-analysis and meta-regression of transcriptomic responses to water stress in Arabidopsis. The Plant Journal, 85 (4), 548-560.

Shaar-Moshe, L., Hübner, S., & Peleg, Z. (2015). Identification of conserved drought-adaptive genes using a cross-species meta-analysis approach. BMC plant biology, 15 (1), 111.

Figures are relevant to the content of the article, but their improvement is necessary to better understand the results. For figures 2 and 5, it would be good to have a higher resolution. However, for Figure 3 this is strictly necessary. Also, figure 3 is desirable to provide a graphic legend of the colors of nodes and links. The perception of the picture is complicated by the presence of a links of different meaning and importance - for example, the "co-expression" for the experimental biologist is treated less unambiguously than the "empirical protein-protein interaction data", for example, and "homology" is of little relevance to the regulation networks. Perhaps it is worth putting out blocks of protein-protein and regulatory interactions into a separate section of the picture.

The work is called "Comprehensive meta-analysis focused on the transcription factors involved in responses to biotic and abiotic stresses in tomato", but in a detailed way it treats the genes associated with a nonspecific response to stress (biotic together with abiotic). In connection with the fact that the authors left out the detailed analysis of all types of stresses - I recommend changing the title, adding the word "non-specific" or "simultaneously" in it.

Experimental design

In the light of current trends in the use of new technologies to improve crop plants, the work looks profitable and certainly contains interesting data. The urgency of the work does not raise any doubts and the topic fits into the journal field.
In this work, the integration of expression data on various stress factors on tomato plants is carried out. For a tomato, such a large-scale work is carried out for the first time and will certainly be of interest to tomato geneticists.
When analyzing transcriptomic data, quite good statistics were used, which makes it possible to unambiguously understand where the strong influence of individual experiments is observed, and where there is a general trend. Methods are described with sufficient information to be reproducible by another investigator.

It should be noted that the results of the GO enrichment analysis revealed the three most common related components - "cellular process", "metabolic process", "response to stimulus". These results are given even in abstract lines 35 - 36. These terms are very broad and not very constructive, at the same time in Table S6 as well as S7 there is a rich list of reliable terms containing more specific terms. Data on GO associated terms should be discussed in detail, possibly below in conjunction with the analysis of associative network.
The results of the analysis of expression experiments and the analysis of GO terms enrichment are supported by sufficient additional materials, which can not be said about the analysis of associative networks.

At the beginning of the network analysis, the number of genes is postulated as "hub role genes", but it is not explained in detail on what basis such a conclusion is drawn. A natural addition to this work is the analysis of node connectivity in the network by various connections separately and together. These data should be welcomed and discussed. Also optional, but recommended This analysis brings a more detailed understanding of the results.

Validity of the findings

The article operates with competent methods of setting tasks, revealing regularities and relies on statistical reliability. References to all used data, just like the main intermediate results are given in the Appendix. The results concerning the found transcription factors are discussed step by step in great detail, which adds to the article a review value and will certainly be of interest to specialists.
Conclusions are clearly formulated and they are justified by the results of the work done.

---

## Round 0.2 · Minor Revisions

Some questions remain without answer - please respond further to Reviewer 3. Please update the figure 3.

Reviewer 3 ·

Basic reporting

The work is still written in fairly good English, excluding ambiguous interpretation. Terms are used correctly and to the place.

Due to the fact that the editorial board decided to send me an article for re-reviewing, I will make some important remarks.
Here and further I will concentrate my consideration on the recommendations that the authors did not follow. I will try to explain in detail what is required to improve the work to the required level.

(1) Strict

Previous recommendation 1:
« I recommend improving the fragment of 77-82 with sufficient details. Meta transcriptome analysis in various variations is actively used for plants before, for example: Rest … (2016) …. Shaar-Moshe …. (2015)»

The authors added the references I mentioned to the article.

Line 79-81
«… and find out some genes that their products are key molecules in the response to stress (Tseng, Ghosh & Feingold, 2012; Shaar-Moshe et al., 2015)»

However, no additional words were given to the description of the results obtained in these papers. It is not clear for what species it is done and what is found. Why is it important? It is important to emphasize the specific value of the analysis that you carried out. This is not seems clearly in the introduction and does not follow from the conclusion. If the authors of the works of 2012 - 2016 managed to find «key molecules in the response to stress» and « figured out interactions between biotic stresses (BS) and abiotic stresses (AS)», then what did you find new?

This article does have scientific value, but it is not visible to the reader without a clear comparison with previous works.

(2) Optional

Previous recommendation 2:
«… For figures 2 and 5, it would be good to have a higher resolution. However, for Figure 3 this is strictly necessary…»

The illustrations are improved and now they are perceived much better. However, the resolution of image 3 is extremely low and may appear careless when printing in a magazine. I recommend to correct it.

Experimental design

(3)
Previous recommendation 3:
«… The results of the analysis of expression experiments and the analysis of GO terms enrichment are supported by sufficient additional materials, which can not be said about the analysis of associative networks.
At the beginning of the network analysis, the number of genes is postulated as "hub role genes", but it is not explained in detail on what basis such a conclusion is drawn. A natural addition to this work is the analysis of node connectivity in the network by various connections separately and together. These data should be included and discussed…»

Still in the text:
200- 201
«Some of these gene possess hub roles in DEGs shared by AS and BS networks (Table S7).»
What is the criterion of choosing the Hub genes?
It would be logical to take a network connectivity into account, but the hubs # 1 and #4 are loosely connected to the network nodes (Fig. 3). This part in results (203-218) and discussion needs to be improved by adding a proper network analysis as recommended before.

Validity of the findings

(3) relates to this part too.

---

## Round 0.3 · accepted · Accept

The reviewers have no more remarks. Since the manuscript has been under review some time, you might be interested in adding recent literature citations from 2018 (you can do so while in production). Transcriptomic studies of biotic and abiotic stress is a fast developing science area.

# Reviewer 3 ·

Basic reporting

The structure of the new version of the article does not have significant weak points.

Experimental design

Acceptable

Validity of the findings

Acceptable